# Effect of Combination Therapy with Probiotic Bulgarian Goat Yoghurt Enriched with *Aronia melanocarpa* Fruit Juice in Patients with Type 2 Diabetes Mellitus and Complication of Diabetic Nephropathy: A Pilot Study

**DOI:** 10.3390/ph18091409

**Published:** 2025-09-19

**Authors:** Petya Goycheva, Kamelia Petkova-Parlapanska, Ekaterina Georgieva, Nikolina Zheleva, Mariya Lazarova, Yanka Karamalakova, Galina Nikolova

**Affiliations:** 1Propaedeutic of Internal Diseases Department, Medical Faculty, Trakia University Hospital, 6000 Stara Zagora, Bulgaria; petya.goycheva@trakia-uni.bg; 2Medical Chemistry and Biochemistry Department, Medical Faculty, Trakia University, 11 Armeiska Str., 6000 Stara Zagora, Bulgaria; kamelia.parlapanska@trakia-uni.bg (K.P.-P.); ekaterina.georgieva@trakia-uni.bg (E.G.); 3Department of “General and Clinical Pathology, Forensic Medicine, Deontology and Dermatovenerology”, Medical Faculty, Trakia University, 11 Armeiska Str., 6000 Stara Zagora, Bulgaria; 4Department of Milk and Dairy Products, Faculty of Agriculture, Trakia University, Students’ Campus, 6000 Stara Zagora, Bulgaria; nikolina.zheleva@trakia-uni.bg (N.Z.); mariya.lazarova@trakia-uni.bg (M.L.)

**Keywords:** diabetes mellitus, diabetic nephropathy, oxidative stress, *Aronia melanocarpa* fruit juice, goat, yoghurt, NO, NOS

## Abstract

**Background**: Goat milk and its fermented products exhibit unique nutritional and therapeutic characteristics corresponding to the management of metabolic disorders, such as type 2 diabetes mellitus (T2DM) and its associated complications, including diabetic nephropathy (DN). Due to its rich content of bioactive compounds and superior digestibility compared to cow’s milk, goat milk enhances nutrient assimilation and exhibits notable anti-inflammatory and antioxidant effects. The primary aim of this investigation was to systematically evaluate the efficacy of goat milk-based nutritional interventions as an integral component of a multifaceted therapeutic approach aimed at attenuating oxidative stress (OS), restoring metabolic homeostasis, and mitigating the progression of long-term complications in patients with diabetes mellitus and concurrent renal dysfunction. **Methods**: Participants diagnosed with T2DM were stratified into three subgroups based on the severity of renal dysfunction. The results were analyzed in comparison with those of healthy control subjects. **Results**: Following a dietary regimen that included goat milk enriched with *Aronia* (*Aronia melanocarpa*) fruit juice patients—particularly those with DN—exhibited marked reductions in free radical concentrations, decreased cytokine production, and diminished levels of lipid and protein oxidation byproducts. Moreover, a significant improvement was observed in nitric oxide (NO) levels, along with partial restoration of the nitric oxide synthase (NOS) system and an upregulation of endogenous antioxidant enzyme activity (*p* < 0.05) relative to pre-intervention measurements. **Conclusions**: These outcomes suggest that the dietary intervention not only attenuated OS but also contributed to improved renal function in affected individuals. The results support the therapeutic potential of functional dairy-based diets—specifically those incorporating bioactive ingredients such as *Aronia melanocarpa* fruit juice and goat milk—in mitigating oxidative damage and enhancing metabolic and renal health in patients with T2DM and DN.

## 1. Introduction

Goat milk has constituted a significant component of the human diet since antiquity, with archeozoological evidence indicating the caprine species was among the first ruminants to be domesticated. Its enduring nutritional relevance is attributed to a dense matrix of essential macro- and micronutrients, comprising high-biological-value proteins, lipids, and minerals. Furthermore, it contains a diverse profile of bioactive constituents, including vitamins, oligosaccharides, polyamines, complex lipids, enzymes, bioactive peptides, and nucleotides [1,2]. These components contribute not only to its basic nutritional value but also to its therapeutic potential. In comparison to cow’s milk, goat milk possesses unique advantages [3]. It contains smaller fat globules and lower levels of αs1-casein, which enhance its digestibility and reduce allergenic potential. Moreover, goat milk facilitates improved absorption of key micronutrients, such as calcium, iron, and copper, particularly in individuals with compromised digestive function or those suffering from malabsorption syndromes. These characteristics make goat milk particularly attractive for use in functional foods and clinical nutrition, especially in vulnerable populations such as infants, the elderly, and individuals with metabolic or inflammatory disorders [4].

Diabetic nephropathy (DN) represents a serious microvascular complication of long-standing type 2 diabetes mellitus (T2DM). Moreover, it is characterized by a gradual decline in kidney function and often progresses silently, without overt symptoms, until advanced stages [5]. Diabetic nephropathy pathogenesis is closely tied to chronic hyperglycemia, which contributes to glucotoxicity and impairs pancreatic β-cell function. Chronic hyperglycemia induces mitochondrial overproduction of reactive oxygen species (ROS) and promotes the formation of reactive nitrogen species (RNS), overwhelming antioxidant defences and causing oxidative stress (OS). Oxidative stress damages lipids, proteins, and DNA, impairs signalling, and drives diabetic complications such as neuropathy, nephropathy, retinopathy, and cardiovascular disease [6,7,8,9]. It also disrupts redox balance, worsens insulin sensitivity, and accelerates vascular and renal injury [10]. The interplay between OS, inflammation, and endothelial dysfunction is central to diabetic nephropathy, where ROS and RNS damage glomerular and tubular cells [11,12]. Moreover, oxidative byproducts with long half-lives serve as reliable biomarkers of oxidative burden in diabetes.

Thus, understanding and targeting oxidative damage is essential for developing effective therapeutic and preventive strategies to mitigate diabetic complications, including DN. In addition to its basic nutritional benefits, goat milk provides a valuable foundation for the production of fermented dairy products such as yoghurt, cheeses, and probiotic beverages. Fermentation is carried out using specific lactic acid bacteria, most notably *Streptococcus thermophilus* and *Lactobacillus delbrueckii* subsp. *bulgaricus*. These bacteria initiate the conversion of lactose into lactic acid and simultaneously generate a range of biologically active peptides and free amino acids, many of which have antioxidant, antihypertensive, and antimicrobial properties [2].

Goat yoghurt, in particular, is a highly nutritious product. It is rich in calcium and potassium and serves as an excellent source of complete proteins, which are vital for muscle maintenance and bone health [1]. Compared to yoghurt derived from cow’s milk, fermented goat milk products typically contain higher levels of essential amino acids, vitamins (particularly B-complex and A), trace elements such as magnesium and zinc, and specific medium-chain fatty acids known for their anti-inflammatory and lipid-lowering properties [13,14]. An important advantage of fermented goat milk is its probiotic potential. The bacterial strains involved in fermentation not only support gut health by modulating intestinal microbiota, but also contribute systemic benefits [15]. These include immune modulation, enhanced antioxidant defence, suppression of low-grade inflammation, improved lipid metabolism, and blood pressure regulation. The ionic form of minerals found in fermented goat products also facilitates better bioavailability and absorption, making them especially beneficial for individuals with increased nutritional demands or impaired nutrient uptake [16]. The natural abundance of calcium, zinc, and magnesium in goat milk also stimulates the proliferation of lactic acid bacteria during the fermentation process, enhancing the functional and sensory qualities of the final product. Recent studies have demonstrated that regular consumption of fermented goat milk can significantly boost the body’s antioxidant capacity, reduce markers of oxidative damage to biomolecules (e.g., lipids and DNA), and improve iron homeostasis [17,18]. These effects are particularly relevant for individuals suffering from anemia or chronic inflammation. The distinct sensory attributes of goat yoghurt—including its smooth, creamy texture, mildly tangy flavour, and characteristic aroma—combined with its multifaceted health benefits, have led to growing consumer interest and increased demand in health-conscious and clinical nutrition markets.

The study investigated the therapeutic potential of probiotic goat yoghurt supplemented with *Aronia melanocarpa* juice as a functional nutritional intervention to mitigate oxidative stress (OS), restore metabolic homeostasis, and reduce the progression of diabetic complications and renal dysfunction. To this end, we assessed (1) free radical generation as an index of OS, (2) oxidative modifications of lipids and proteins, (3) the activity of key antioxidant enzymes, and (4) cytokine profiles, with a focus on interleukin expression.

## 2. Results

### 2.1. Goat Yoghurt with Aronia

Goat’s yoghurt fortified with *Aronia* juice offers notable nutritional benefits, particularly as a natural source of vital macro- and microelements. This combination enhances the overall mineral profile of the yoghurt, making it a valuable addition to a balanced diet. The inclusion of *Aronia* juice not only contributes bioactive compounds like antioxidants but also increases the content of essential minerals such as calcium, potassium, magnesium (macroelements), and trace elements like zinc, iron, and manganese (microelements) [2]. Regular consumption of this enriched yoghurt can help support a range of physiological processes necessary for maintaining health, including bone metabolism, enzyme activity, immune function, and antioxidant defence. Due to its natural composition and enhanced mineral content, this product can be considered a functional food that contributes positively to nutritional intake and may help in preventing mineral deficiencies in the diet.

### 2.2. NO, eNOS and iNOS Assay

In the cohort with preserved renal function (DMT2N0, stage 1), NO radical levels were significantly elevated relative to healthy controls (*p* = 0.05) (Figure 1, NO) and to patients with DN at stages 3a (*p* = 0.05) and 3b (*p* = 0.0003). Conversely, endothelial nitric oxide synthase (eNOS) and inducible nitric oxide synthase (iNOS) concentrations (Figure 1, eNOS; Figure 1, iNOS) were markedly reduced in groups with impaired renal function compared to stage 1 patients. Following a three-month intervention with daily consumption of 200 g of goat yoghurt enriched with Aronia and adherence to a standardized diet, significant alterations were observed in patients with renal impairment. In the DMT2N0 group with normal renal function, post-intervention analysis revealed a significant reduction in NO radicals (*p* < 0.05) compared to baseline values. Importantly, in cohorts with renal dysfunction, baseline depletion of NO, eNOS, and iNOS was followed by partial restoration of these parameters after the dietary intervention.

### 2.3. Measuring the Levels of Malondialdehyde (MDA), ROS, and 8-Iso-Prostaglandins (8-IsoPr)

Significantly elevated levels of MDA were observed in both DN groups—those with mild to moderate renal impairment (*p* = 0.05) and those with moderate to severe impairment (*p* = 0.02)—when compared to the DMT2N0 group with preserved renal function. Furthermore, all three patient groups demonstrated significantly higher MDA concentrations relative to healthy controls (*p* = 0.002, Figure 2, MDA). These findings indicate a marked increase in OS associated with both diabetes and declining renal function. Notably, a specialized diet incorporating goat milk and *Aronia* juice resulted in significant modulation of MDA levels, suggesting a potential therapeutic effect of the dietary intervention on oxidative stress biomarkers.

Consistently, the nephropathy cohorts exhibited a pronounced elevation in ROS and 8-iso-prostaglandin F_2_α (8-Iso-PGF), a robust biomarker of lipid peroxidation, when compared with both the diabetes-only group and normoglycemic controls (Figure 2, ROS; 2, 8-Iso-PGF). Following dietary supplementation with goat milk and *Aronia* juice, a significant reduction in ROS and 8-Iso-PGF levels was observed in the DN groups, compared to their respective pre-intervention values and to DN groups not receiving the diet. These results further support the antioxidant potential of the dietary intervention in mitigating OS in patients with DN.

Correlation analysis demonstrated a strong inverse association between plasma MDA levels and eGFR (r = −0.98, *p* ≤ 0.01), alongside a positive correlation with UAE (r = 0.80, *p* ≤ 0.03). Likewise, plasma 8-Iso-PGF exhibited a significant negative correlation with eGFR (r = −0.97, *p* ≤ 0.04) and a positive correlation with UAE (r = 0.92, *p* ≤ 0.04) following yoghurt supplementation.

### 2.4. Levels of Protein Modification

Protein modifications were assessed through the quantification of protein carbonyls (PC), 3-maleimido-PROXYL (5-MSL), and advanced glycation end products (AGEs). Relative to healthy controls, PC concentrations (Figure 3, PC) were significantly elevated across all diabetic subgroups, both with and without complications: DMT2N0 stage 1 (*p* = 0.001), DMT2N1 stage 3a (*p* = 0.002), and DMT2N2 stage 3b (*p* = 0.003). Among patients with renal impairment, PC values were comparable across groups and exhibited non-significant increases relative to the DMT2N0 stage 1 cohort (*p* = 0.002).

The levels of 5-MSL were significantly elevated in all diabetic subgroups compared to healthy controls: DMT2N0 stage 1 (*p* = 0.05), DMT2N1 stage 3a (*p* = 0.03), and DMT2N2 stage 3b (*p* = 0.02). Following a three-month dietary intervention with goat milk yogurt enriched with *Aronia melanocarpa* juice, 5-MSL concentrations were significantly reduced in the intervention group relative to the corresponding non-intervention group (*p* < 0.05), suggesting a potential antioxidative effect of the regimen on protein thiol oxidation (Figure 3, 5-MSL). In the DMT2N0 group, advanced glycation end products (AGEs) were significantly elevated compared to controls (*p* = 0.01; Figure 3, AGEs). Among patients with impaired renal function, AGE levels were further increased relative to controls (DMT2N1 stage 3a, *p* = 0.01; DMT2N2 stage 3b, *p* = 0.03) and showed significant differences when compared to the DMT2N0 stage 1 group (DMT2N1 stage 3a, *p* = 0.01; DMT2N2 stage 3b, *p* = 0.02).

Correlation analysis revealed a strong inverse relationship between serum PC and IL-10 (r = −0.92, *p* ≤ 0.02) and between AGEs and IL-10 (r = −0.94, *p* ≤ 0.001). Additionally, positive associations were observed between serum NO and eNOS (r = 0.90, *p* ≤ 0.01) as well as NO and iNOS (r = 0.93, *p* ≤ 0.01).

### 2.5. Antioxidant Enzymes

Analysis of the data revealed that, in patients with DN, the activities of key antioxidant enzymes—superoxide dismutase (SOD; Figure 4), catalase (CAT; Figure 4), and glutathione peroxidase (GSH-Px; Figure 4)—were significantly diminished in cohorts with impaired renal function prior to dietary intervention (*p* < 0.05) relative to both healthy controls and diabetic patients with preserved renal function. Following a three-month regimen of goat yoghurt supplemented with Aronia melanocarpa juice, SOD activity was significantly elevated in the DMT2N1 stage 3a group compared to baseline (*p* = 0.01; Figure 4, SOD), with a similar increase observed in the DMT2N2 cohort (*p* = 0.05), indicating a restorative effect of the dietary intervention on enzymatic antioxidant defences.

There was a statistically significant difference in CAT activity between the groups with DN, DMT2N1 (*p* = 0.004) and DMT2N2 (*p* = 0.002), and patients with diabetes without complications, DMT2N0 (Figure 4, CAT), compared to controls. Also, CAT activity was statistically significantly higher in DMT2N1 (*p* = 0.01) and DMT2N2 (*p* = 0.01) after yoghurt intake compared to the results in the groups before the diet. Analysis of the results showed that both groups, DMT2N1 (0.01) and DMT2N2 (0.01), had significantly lower GHS-Px activity before therapy (Figure 4) compared to DMT2N0 and controls (*p* = 0.01; Figure 4, GHS-Px). After yoghurt intake, GHS-Px activity was statistically significantly increased. Analysis of the results after intake of goat yoghurt showed a statistically significant improvement in the groups with DN compared to the same groups before intake (*p* < 0.05; Figure 4).

### 2.6. Pro-Inflammatory Cytokine Levels

Results before the goat yoghurt and aronia diet: in both groups of patients with DMT2N1 and DMT2N2, which had different percentages of kidney function loss, interleukin 10 (IL-10) levels (Figure 5) were significantly lower compared to both DMT2N0 with normal kidney function (*p* < 0.05) and the control group (*p* < 0.05). Interleukin 18 (IL-18) levels in the DMT2N0 stage 1 group were significantly higher compared to DMT2N1 stage 3a (*p* < 0.05) and DMT2N2 stage 3b (*p* < 0.05). A similar significant relationship was observed for interleukin 6 (IL-6). The mean tumour necrosis factor (TNF-α; Figure 5) levels in the DN groups were significantly higher compared to the control groups (*p* < 0.05) and DMT2N0 (*p* < 0.05). The results for interferon gamma (IFN-γ; Figure 5) showed a significant increase in groups with loss of renal function compared to controls (DMT2N1 stage 3a, *p* < 0.05; DMT2N2 stage 3b, *p* < 0.05) and DMT2N0 stage 1 (*p* < 0.05). Transforming growth factor-β (TGF-β; Figure 5) was significantly higher in groups with impaired renal function compared to healthy volunteers (DMT2N1 stage 3a, *p* < 0.05; DMT2N2 stage 3b, *p* < 0.05) and DMT2N0 stage 1 (*p* < 0.05). Analysis of the results after the goat yoghurt and aronia diet showed a significant increase, especially in groups with kidney function loss (DMT2N1 stage 3a, *p* < 0.05; DMT2N2 stage 3b, *p* < 0.05), compared to both the results before therapy and the controls.

## 3. Discussion

OS is a complex phenomenon, largely due to the dual role of ROS in regulating both physiological and pathological processes. It arises from an imbalance in the redox state, characterized by an excessive production of ROS that overwhelms the body’s antioxidant defence systems, leading to cellular and molecular damage [19]. The redox homeostatic system plays a critical role in enabling cells and tissues to respond to environmental and metabolic stressors. This is mediated through a series of tightly regulated redox reactions that maintain the equilibrium between reversible and irreversible oxidation of key molecular targets, such as transcription factors [20]. Among ROS, non-radical species like hydrogen peroxide (H_2_O_2_) and singlet oxygen are generated in a spatially and temporally controlled manner, allowing them to act as important secondary messengers in redox signaling pathways. H_2_O_2_, in particular, is involved in numerous biological processes including insulin signaling, cell proliferation, differentiation, immune responses, and tissue repair [21]. At physiological concentrations, H_2_O_2_ and singlet oxygen act as signaling molecules that, in coordination with redox-sensitive transcription factors such as nuclear factor erythroid 2–related factor 2 (Nrf2), help fine-tune intracellular redox homeostasis and support cellular adaptation to mild OS. This adaptive response is referred to as oxidative eustress [22]. However, when H_2_O_2_ levels rise beyond the physiological threshold—becoming supraphysiological—they induce oxidative damage to cellular macromolecules, including proteins, lipids, and nucleic acids. This state is defined as OS and is implicated in the pathogenesis of various chronic and degenerative diseases [23].

Diabetes initiates a range of pathophysiological processes that, depending on the stage of the disease, can either elevate or reduce the NO production and its metabolites. Increased levels of NO metabolites in plasma have been observed in individuals with either normal or mildly impaired renal function, as well as in those with DN. Notably, in diabetic patients with preserved renal function, NO level were significantly elevated, while nitrite/nitrate (NO_2_^−^/NO_3_^−^) concentrations are markedly reduced [24]. The development of DN is associated with alterations in the eNOS expression and activity [25]. Several studies have indicated that renal eNOS expression and activity were upregulated during the early stages of DN but become downregulated in the disease chronic phase. The marked increase in NO levels in the early stages is believed to contribute to microalbuminuria, a condition primarily mediated by both eNOS and neuronal nitric oxide synthase (nNOS) [26,27]. In the renal vasculature, eNOS is the predominant enzyme, and its expression is heightened during the initial stages of renal injury due to hyperglycemia [24,28].

In the present study, was confirmed that patients with diabetes but normal renal function exhibited elevated NO radicals and eNOS levels. Conversely, in individuals with moderate to severe renal dysfunction, NO and eNOS levels were significantly diminished. Notably, a diet incorporating goat yoghurt and Aronia demonstrated a statistically significant reduction in the NO and eNOS levels in patients with T2DM and normal renal function, while there was a corresponding increase in NO and eNOS levels in patients with DN (Figure 1). Normally, in DN, there is a pathological imbalance between the protective effects of eNOS and the harmful effects of iNOS. Under normal physiological conditions, eNOS-derived NO maintains vascular homeostasis by promoting vasodilation, inhibiting platelet aggregation, and suppressing inflammation. However, in the diabetic milieu, chronic hyperglycemia and OS down regulate eNOS expression and activity, leading to reduced physiological NO production and endothelial dysfunction. Conversely, hyperglycemia and pro-inflammatory cytokines, such as TNF-α, IL-1, and IL-6, upregulate iNOS expression, resulting in excessive and sustained NO production. Unlike the tightly regulated NO from eNOS, high-output NO from iNOS can react with superoxide to form peroxynitrite (NOO^−^), a potent reactive nitrogen species. This contributes to oxidative and nitrosative stress, promoting inflammation, apoptosis, and extracellular matrix accumulation—key drivers of glomerular and tubulointerstitial damage in DN [29].

The combination of a goat yoghurt-based diet with *Aronia* juice was shown to improve fasting glucose levels and glucose tolerance, increase NO levels in individuals with impaired kidney function (T2DMN1 and T2DMN2), and reduce these levels close to the controls in T2DMN0 with normal renal function. Based on the significant changes observed in various metabolites, key metabolic pathways, activities of antioxidant enzymes, and biochemical reactions can be identified as critical integrated processes involved. These findings shed light on the metabolism of lactobacilli during fermentation when combined with chokeberry fruit juice addition. Hyperglycemia-induced OS triggers the production of inflammatory cytokines (Figure 5). Elevated blood sugar levels lead to increased circulating levels of cytokines such as IL-6, TNF-α, and IL-18, which may contribute to impaired glucose tolerance even in healthy individuals through oxidative mechanisms. In particular, IL-6 levels are independently correlated with C-reactive protein (CRP) in adults with type II diabetes, indicating a link between inflammation and metabolic dysfunction.

In diabetes mellitus, chronic hyperglycemia leads to ROS and RNS overproduction. For example, high glucose levels increase metabolic flux through glycolysis and the tricarboxylic acid cycle, which generates an increase in NADH and FADH_2_ levels. This leads to increased electron transport activity in mitochondria, overloading the mitochondrial electron transport chain and electron leakage. From premature reaction with molecular oxygen, superoxide anion radical (O_2_•^−^) is formed, which is the main form of ROS. Hyperglycemia activates protein kinase C, which stimulates the enzyme NADPH oxidase, responsible for the transfer of electrons from NADPH to oxygen, which additionally produces superoxide radicals. In the polyol pathway, glucose is reduced to sorbitol by the enzyme aldose reductase, which uses NADPH. NADPH is involved in the regeneration of the antioxidant glutathione (GSH), and its depletion reduces the antioxidant capacity of the cell, resulting in increased OS [30]. High glucose levels are responsible for increasing the levels of AGEs. These molecules interact with their receptor RAGE on the cell surface, triggering signaling cascades such as activation of NF-κB, which enhances the production of pro-inflammatory cytokines and ROS. Last but not least, there is the influence of eNOS. In diabetic conditions, eNOS is uncoupled, as a result of oxidative damage and reduced levels of the cofactor tetrahydrobiopterin. In the uncoupling process, eNOS produces superoxide anion radical instead of nitric oxide (NO), as a result of the reaction between O_2_•^−^ and NO, ONOO^−^, a highly reactive nitrogen species, is produced, and nitrosative stress is increased. Therefore, hyperglycemia induces an increase in ROS and RNS production through mitochondrial dysfunction, activation of NADPH oxidase, antioxidant depletion, AGE-RAGE signaling, and uncoupling of eNOS.

These processes contribute significantly to oxidative and nitrosative stress, which are fundamental to the pathogenesis of diabetes and its complications (Figure 1).

When combined, goat yoghurt may act as a carrier that enhances the bioavailability and gastrointestinal stability of Aronia polyphenols, allowing for more efficient absorption and systemic antioxidant effects. The synergistic action of bioactive compounds in both components likely contributes to the significant reduction in oxidative stress biomarkers observed in the study, such as 5-MSL intensity of free nitroxide radical, PC, and AGEs. A study of the effect of three daily doses of 50 mL of chokeberry juice, corresponding to 258 mg of anthocyanins, for three months found a significant reduction in LDL cholesterol after taking chokeberry supplements, but only a minor reduction in fasting blood sugar, hemoglobin A1c (HbA1c), total cholesterol, and triglycerides [31]. At a higher dose (200 mL) of aronia juice twice daily for three months, a significant reduction in fasting blood sugar, HbA1c and cholesterol was reported, along with a slight improvement in blood pressure [32]. In healthy subjects, a significant reduction in postprandial blood sugar was found when 100 mL of aronia juice was administered before meals [33].

Anthocyanins are natural water-soluble pigments, belong to the group of flavonoids and are powerful antioxidants. Aronia is the absolute record holder in anthocyanin content among all widely distributed berries. Anthocyanins in Aronia are extremely powerful and effective. The results demonstrate that administering goat yoghurt together with chokeberry juice significantly reduces lipid peroxidation and free radical levels, (see Figure 2, ROS). CRP is a major acute-phase protein primarily produced and secreted by liver cells in response to cytokines like IL-6 during inflammation. Elevated levels of pro-inflammatory cytokines, especially IL-6, along with CRP and other markers, are common targets for intervention in managing type 2 diabetes.

Usually, IL-6 and IL-10 show opposite expression patterns due to their different roles in inflammation. IL-6 is a pro-inflammatory cytokine, and its levels are often elevated in response to chronic low-grade inflammation, contributing to insulin resistance and disease progression [34]. In contrast, IL-10 is an anti-inflammatory cytokine that normally suppresses cytokines like IL-6. However, IL-10 levels are often reduced in diabetes, leading to uncontrolled inflammation. This imbalance reflects a shift toward a pro-inflammatory state that worsens metabolic dysfunction [35].

In the investigation of oxidative stress-related mechanisms underlying DN, a multifaceted approach was employed, combining site-specific spin labeling, protein oxidation markers, and glycation end-product analysis. A nitroxide-based spin label 5-MSL, was used in conjunction with EPR spectroscopy to assess conformational dynamics and microenvironmental changes in thiol-containing proteins such as albumin [30]. This allowed for the detection of altered protein mobility and structural rigidity associated with oxidative damage in DN. In parallel, protein carbonyl content—a well-established biomarker of irreversible protein oxidation—was quantified to provide a broader measure of oxidative protein damage [36]. Additionally, AGEs were measured to evaluate the extent of non-enzymatic glycation and oxidative cross-linking of proteins, processes that are known to be exacerbated in hyperglycemic and pro-oxidative states such as DN [37]. The combined analysis of these parameters offers a comprehensive understanding of oxidative and carbonyl stress in the progression of DN, linking structural protein modifications to functional impairments in renal tissue. The dietary intervention consisting of goat milk and Aronia juice led to a statistically significant reduction in oxidative and carbonyl stress markers across patients with varying degrees of renal impairment. Specifically, post-intervention levels of 5-MSL-labeled protein mobility, protein carbonyls, and AGEs were significantly decreased compared to pre-intervention values, indicating improved redox status and reduced protein oxidative modification. These findings suggest that the combined dietary components exert a protective effect against oxidative damage in patients with DN.

Goat yoghurt is notably rich in leucine, an amino acid that plays a vital role in stimulating insulin secretion from pancreatic β-cells. Leucine also helps enhance glucose and insulin sensitivity. Several studies have highlighted that goat yoghurt contains a high level of oligosaccharides, which possess prebiotic effects, and anti-inflammatory properties. Furthermore, chokeberry (*Aronia Melanocarpa*) flavonoids reduced CRP levels in the blood, thereby lowering pro-inflammatory cytokine production. Flavonoids are powerful antioxidants capable of reducing the formation of free radicals by breaking down hydrogen peroxide without generating additional radicals. For example, consuming soy germ enriched with isoflavones has been shown to improve overall plasma antioxidant capacity and glutathione levels in patients with type 2 diabetes [38]. Diets rich in flavonoids have been associated with decreased levels of CRP and lipid peroxidation in individuals with type 2 diabetes and nephropathy. Long-term intake of yoghurt proteins has also been linked to reductions in CRP, blood glucose, total cholesterol, LDL cholesterol, and triglycerides among patients with type 2 diabetes and nephropathy. The decrease in cytokines and lipid oxidation products observed in diabetic patients consuming goat yoghurt suggests a reduction in OS, likely attributable to the high content of oligosaccharides in goat milk. Overall, these findings suggest that diets incorporating goat yoghurt and chokeberry juice could be effective in mitigating inflammation and OS associated with diabetes, contributing to improved metabolic health.

## 4. Materials and Methods

### 4.1. Source of Raw Materials

The goat yoghurt used in this study was supplied by a student farm and is subject to seasonal fluctuations. The breed is Bulgarian White Dairy Goat. Lactation in the goats occurs over an average duration of 305 days per year, typically spanning from March-April to November-December. Goats are generally inseminated in late August to early September, with offspring born in January and February. Following a suckling period of approximately 60 days, the animals were milked twice daily using machine-assisted techniques—once in the morning and once in the evening. For this study, the milk used to produce the experimental yoghurt containing *Aronia melanocarpa* was collected as pooled milk from the morning milking session during the early- to mid-lactation period, specifically from March to July. The yoghurt was prepared immediately after sample collection, ensuring the final product was ready within 24 h. Due to the limited quantity of yoghurt produced, the study’s sample size was constrained, which consequently restricted the number of participants included in the research. *Aronia melanocarpa* juice is 100% organic juice made by cold-pressing and pasteurizing fresh fruits from the certified local retailer to ensure quality and traceability.

### 4.2. Yoghurt Preparation Protocol

Yoghurt production was carried out in the laboratory of the “Milk and Dairy Products Technology” section, following traditional fermentation techniques. Initially, the raw goat milk underwent pasteurization by heating to 90 °C and maintaining that temperature for 30 min to ensure microbial safety and deactivate native enzymes. After pasteurization, the milk was cooled to 46 °C to create optimal conditions for fermentation. At this stage, the milk was inoculated with a 1.5% starter culture comprising *Lactobacillus delbrueckii* subsp. *bulgaricus* and *Streptococcus thermophilus* (commercially available as Laktina 17, Bankya, Sofia, Bulgaria). The inoculated milk was enriched with 3% cold-pressed Aronia (*Aronia Melanocarpa*) juice, thoroughly mixed to ensure even distribution. The prepared mixtures were dispensed into 200 g sterile containers and incubated (thermostated) at 43 °C until coagulation was achieved. Once fermentation was complete, the yoghurt was immediately cooled to 10 °C to halt microbial activity and then stored under refrigeration conditions at 4–6 °C until further analysis.

### 4.3. Chemicals

All reagents were analytically grade and were purchased from Merck Bulgaria EAD (Sofia, Bulgaria).

### 4.4. Subjects and Study Design

This study enrolled a total of 122 individuals diagnosed with T2DM, with a male-to-female ratio of approximately 1:1.5. The demographic diversity of the study cohort reflects the broad clinical spectrum of the T2DM population (Table 1). Notably, 59.4% of the participants were classified as obese, highlighting the high prevalence of obesity as comorbidity within the group. Participants with T2DM were stratified into three groups based on the severity of DN, as determined by their estimated glomerular filtration rate (eGFR) and urinary albumin excretion (UAE) values. The clinical and laboratory characteristics of these groups were then compared with a control group consisting of 42 age-matched, healthy volunteers. Venous blood samples were collected from all participants in the early morning following an overnight fast, specifically for the purpose of lipid profile analysis. Blood intended for EPR analysis was processed immediately upon collection. The remaining samples were stored at −80 °C for subsequent enzyme-linked immunosorbent assay (ELISA) testing. To assess glycaemic control, fasting plasma glucose (FPG) and glycated hemoglobin (HbA1c) levels were measured. HbA1c served as an indicator of long-term glycaemic exposure. The study was conducted following the ethical standards of the institutional research committee of the Medical Faculty and Hospital and adhered to the ethical principles outlined in the 1964 Declaration of Helsinki and its subsequent amendments or equivalent ethical guidelines.

At baseline, 54 (58%) of patients with type 2 diabetes mellitus (T2DM) exhibited inadequate glycemic control, defined as HbA1c > 7% and fasting plasma glucose > 6.1 mmol/L (Table 1). Regarding pharmacotherapy, T2DM patients with good glycemic control (n = 24) were treated with oral hypoglycemic agents, including sulfonylureas and biguanides. Among diabetic nephropathy (DN) patients with mild to moderate renal impairment (eGFR stage 3a; n = 47), 24 received sulfonylureas and biguanides, while the remaining 23 were treated with sulfonylureas in combination with basal insulin. All patients with moderate to severe renal impairment (eGFR stage 3b; n = 52) were on a basal-bolus insulin regimen, comprising preprandial rapid-acting insulin and an evening basal dose. Comparisons were made with 42 healthy volunteers, of whom only 7 (7.4%) presented with modest overweight (BMI 31–33). Study participants were matched for gender. Subsequently, all patients were administered 200 g of goat yogurt enriched with *Aronia melanocarpa* juice daily for three months in conjunction with a controlled diet. Table 2 summarizes post-intervention outcomes.

### 4.5. Electron Paramagnetic Resonance (EPR) Spectroscopy

All EPR measurements were performed at room temperature on a Bruker, BioSpin GmbH, Ettlingen Germany, equipped with standard Resonator. All EPR experiments were carried out in triplicate and repeated thrice. Spectral processing was performed using Bruker, WIN-EPR version 2021 and SimFonia software version 2021.

#### 4.5.1. Evaluation the ROS Products Levels

The levels of ROS were determined according to Shi et al. [39] with some modification.

#### 4.5.2. Evaluation of NO Levels

Based on the methods published by Yoshioka et al. [40] and Yokoyama et al. [41] we developed and adapted the EPR method for estimating the levels of NO radicals in plasma.

#### 4.5.3. Protein Oxidation with 3-Maleimido Proxyl (5-MSL)

The extent of protein and albumin impairment in the renal samples was evaluated through the in vivo EPR technique utilizing spin conjugation with 5-MSL [42].

### 4.6. Enzyme-Linked Immunosorbent Assay

All oxidative stress (OS) markers were quantified using commercially available ELISA kits in accordance with the manufacturers’ protocols.

### 4.7. Limitations of the Study

A primary limitation of this study is the absence of a placebo-controlled group. Leveraging the known antioxidant and anti-inflammatory properties of *Aronia melanocarpa*, the study focused on patients with type 2 diabetes and diabetic nephropathy, in whom oxidative stress is a key pathogenic factor. The aim was to evaluate goat yogurt–based dietary interventions for mitigating OS, restoring metabolic balance, and reducing long-term complications, with comparative analyses across subgroups stratified by nephropathy status and KDIGO 2022 renal function stages.

Additional limitations include the cross-sectional design, which precludes assessment of temporal or causal relationships, and the exclusion of patients with acute decompensation or advanced renal failure (eGFR < 30 mL/min/1.73 m^2^ or on hemodialysis), limiting generalizability. Renal function and albuminuria were assessed at a single time point, potentially failing to capture dynamic changes. Moreover, unmeasured confounders, including medication, comorbidities, and lifestyle factors, may have influenced the observed associations.

### 4.8. Statistical Analysis

Statistical analyses were conducted using Statistica 8 (StaSoft, Inc., Tulsa, OK, USA). Data are presented as means ± standard error (S.E.). Group differences were assessed by one-way ANOVA, followed by Fisher’s least significant difference (LSD) post hoc test to identify specific intergroup differences, with *p* < 0.05 considered statistically significant. Figures were generated using GraphPad Prism 10 for Windows 64-bit, Version 10.6.0 (2025).

## 5. Conclusions

Oxidative stress plays a dualistic role in the body, functioning both as a key regulatory mechanism and a driver of pathological conditions when redox balance is disrupted. In diabetes, particularly DN, redox imbalance and altered NO signaling are central to disease progression. The current study confirms that NO and eNOS levels vary according to renal function status in diabetic individuals, with elevated levels observed in early-stage DN and reductions in advanced disease. Importantly, the introduction of a diet enriched with goat yoghurt and chokeberry (*Aronia Melanocarpa*) juice demonstrated promising results in modulating OS and inflammation. This dietary intervention not only normalized NO and eNOS levels according to renal function but also reduced key inflammatory markers, lipid peroxidation, and blood glucose levels. The observed benefits are likely attributed to the antioxidant properties of flavonoids in *Aronia* and the anti-inflammatory, insulin-sensitizing effects of goat yoghurt components such as leucine and micro- and macro- elements. These findings highlight the potential of functional nutrition in managing OS and inflammation in type 2 diabetes and its complications. Future research should explore these dietary strategies further as accessible, adjunctive therapies in diabetes care.

## Data Availability

The original contributions presented in this study are included in the article. Further inquiries can be directed to the corresponding author.

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
