# Peer review of "Effect of Combination Therapy with Probiotic Bulgarian Goat Yoghurt Enriched with Aronia melanocarpa Fruit Juice in Patients with Type 2 Diabetes Mellitus and Complication of Diabetic Nephropathy: A Pilot Study"

_pharmaceuticals, 2025, doi:10.3390/ph18091409_

Round 1
Reviewer 1 Report
Comments and Suggestions for Authors
The article "Effect of combination therapy with probiotic Bulgarian goat yogurt enriched with Aronia melanocarpa fruit juice in patients with type 2 diabetes mellitus and complications of diabetic nephropathy: A Pilot Study" presents very interesting results. The introduction presents adequate and up-to-date information on the topic. The results and procedures are consistent with the objectives. However, I have several comments.
1. How did you choose the sample size?
2. Please include the type of study in the Materials and Methods section.
3. What limitations does the study have?
4. Do the results suggest a pharmacokinetic influence due to the active ingredients in the combination?
5. Improve the quality of the figures.
6. Mark the significant differences between the variables analyzed in the tables.
7. What does the LSD post hoc test mean?
Author Response
Dear Reviewer,
Thank you for your valuable comment and for helping us improves our manuscript.
Please find the corrected passages in red in our manuscript
Reviewer 1
The article "Effect of combination therapy with probiotic Bulgarian goat yogurt enriched with Aronia melanocarpa fruit juice in patients with type 2 diabetes mellitus and complications of diabetic nephropathy: A Pilot Study" presents very interesting results. The introduction presents adequate and up-to-date information on the topic. The results and procedures are consistent with the objectives. However, I have several comments.
Point 1. How did you choose the sample size?
Response 1: This study is part of a broader research initiative aimed at evaluating the antioxidant properties of goat yoghurt enriched with Aronia melanocarpa. The goat milk utilised in this research was sourced from a student training farm and is subject to seasonal variations. The breed is Bulgarian White Dairy Goat. Lactation in the goats occurs over an average duration of 305 days per year, typically spanning from March-April to November-December. Goats are generally inseminated in late August to early September, with offspring born in January and February. Following a suckling period of approximately 60 days, the animals are milked twice daily using machine-assisted techniques—once in the morning and once in the evening. For this study, the milk used to produce the experimental yoghurt containing Aronia melanocarpa was collected as pooled milk from the morning milking session during the early- to mid-lactation period, specifically from March to July. The yoghurt was prepared immediately after sample collection, ensuring the final product was ready within 24 hours. Due to the limited quantity of yoghurt produced, the study's sample size was constrained, which consequently restricted the number of participants included in the research. Aronia melanocarpa juice is 100% organic juice made by cold-pressing and pasteurising fresh fruits from the company "Aronia". That why we also not use the placebo group.
Point 2. Please include the type of study in the Materials and Methods section.
Response 2: This is a prospective, follow-up study.
Point 3. What limitations does the study have?
Response 3: This study presents several limitations that must be considered when interpreting the findings. First, the cross-sectional design employed in this research does not permit the assessment of temporal or causal relationships between the decline in renal function, oxidative stress, inflammation, and metabolic parameters. Second, the exclusion of patients experiencing acute decompensation, such as those with diabetic ketoacidosis, as well as individuals with advanced renal failure (eGFR < 30 mL/min/1.73 m² or those undergoing hemodialysis), restricts the generalizability of the results to patients with more severe or unstable conditions. Third, the evaluation of renal function and albuminuria was conducted at a single time point, which may not adequately represent the dynamic changes that occur over time. Furthermore, potential confounding factors, including medication use, comorbidities, and lifestyle variables, were not thoroughly controlled for, potentially influencing the observed associations. All this is described in detail in the part "Material and methods".
Point 4. Do the results suggest a pharmacokinetic influence due to the active ingredients in the combination?
Response 4: In our study, we did not investigate pharmacokinetic parameters, but only biochemical and oxidative stress, the analysis of the results shows that there is a statistical difference between the diet and yogurt groups. In the next stage, we will follow up on the pharmacokinetic parameters.
Point 5. Improve the quality of the figures.
Response 5: Done
Point 6. Mark the significant differences between the variables analyzed in the tables.
Response 6: Done
Point 7. What does the LSD post hoc test mean?
Response 7: The Least Significant Difference (LSD) test is a post hoc statistical test used after an Analysis of Variance (ANOVA) finds a significant overall difference among multiple groups to identify which specific pairs of group means are statistically different from each other. It uses a t-test approach to calculate a minimum difference (the "least significant difference") between any two means. If the actual difference between two means is greater than this calculated LSD, then those two group means are considered statistically different.
Reviewer 2 Report
Comments and Suggestions for Authors
---The authors state that “Prolonged exposure to elevated glucose levels promotes the excessive generation of reactive oxygen species (ROS) and reactive nitrogen species (RNS), leading to a state of oxidative stress (OS).” Please provide appropriate references to support this statement. In addition, it would be helpful to explain the underlying mechanisms by which elevated glucose levels induce the production of ROS and RNS, as this would improve the reader’s understanding.
---In Figure 3, the font size of the x- and y-axis labels should be increased, as they are currently difficult to read. Furthermore, the authors should discuss the fundamental reason why IL-10 and IL-6 exhibit opposite expression trends.
---The manuscript mentions that goat milk enriched with Aronia regulates free radical concentrations. Could the authors elaborate on the underlying mechanism of this effect? Such as a schematic diagram would further clarify this point for readers.
Author Response
Dear Reviewer,
Thank you for your valuable comment and for helping us improves our manuscript.
Please find the corrected passages highlighted in yellow
Point 1: The authors state that “Prolonged exposure to elevated glucose levels promotes the excessive generation of reactive oxygen species (ROS) and reactive nitrogen species (RNS), leading to a state of oxidative stress (OS).” Please provide appropriate references to support this statement.
Response 1: done
Point 2: In addition, it would be helpful to explain the underlying mechanisms by which elevated glucose levels induce the production of ROS and RNS, as this would improve the reader’s understanding.
Response 2: In diabetes mellitus, chronic hyperglycemia leads to ROS and RNS overproduction. For example, high glucose levels increase metabolic flux through glycolysis and the tricarboxylic acid cycle, which generates an increase in NADH and FADH₂ levels. This leads to increased electron transport activity in mitochondria, overloading the mitochondrial electron transport chain and electron leakage. From premature reaction with molecular oxygen, superoxide anion radical (O₂⁻) is formed, which is the main form of ROS. Hyperglycemia activates protein kinase C, which stimulates the enzyme NADPH oxidase, responsible for the transfer of electrons from NADPH to oxygen, which additionally produces superoxide radicals. In the polyol pathway, glucose is reduced to sorbitol by the enzyme aldose reductase, which uses NADPH. NADPH is involved in the regeneration of the antioxidant glutathione (GSH), and its depletion reduces the antioxidant capacity of the cell, resulting in increased oxidative stress. High glucose levels are responsible for increasing the levels of advanced glycation end products (AGEs). These molecules interact with their receptor (RAGE) on the cell surface, triggering signaling cascades such as activation of NF-κB, which enhances the production of pro-inflammatory cytokines and ROS. Last but not least, there is the influence of endothelial nitric oxide synthase (eNOS). In diabetic conditions, eNOS is uncoupled, as a result of oxidative damage and reduced levels of the cofactor tetrahydrobiopterin. In the uncoupling process, eNOS produces superoxide anion radical instead of nitric oxide (NO), as a result of the reaction between O₂⁻ and NO, peroxynitrite (ONOO⁻), a highly reactive nitrogen species, is produced, and nitrosative stress is increased. Therefore, hyperglycemia induces an increase in ROS and RNS production through mitochondrial dysfunction, activation of NADPH oxidase, antioxidant depletion, AGE-RAGE signaling, and uncoupling of eNOS. These processes contribute significantly to oxidative and nitrosative stress, which are fundamental to the pathogenesis of diabetes and its complications.
Point 3: In Figure 3, the font size of the x- and y-axis labels should be increased, as they are currently difficult to read.
Response 3: Done
Point 4: Furthermore, the authors should discuss the fundamental reason why IL-10 and IL-6 exhibit opposite expression trends.
Response 4: In diabetes mellitus, IL-6 and IL-10 show opposite expression patterns due to their different roles in inflammation. IL-6 is a pro-inflammatory cytokine, and its levels are often elevated in response to chronic low-grade inflammation, contributing to insulin resistance and disease progression. In contrast, IL-10 is an anti-inflammatory cytokine that normally suppresses cytokines like IL-6. However, IL-10 levels are often reduced in diabetes, leading to uncontrolled inflammation. This imbalance reflects a shift toward a pro-inflammatory state that worsens metabolic dysfunction.
Point 5: The manuscript mentions that goat milk enriched with Aronia regulates free radical concentrations. Could the authors elaborate on the underlying mechanism of this effect? Such as a schematic diagram would further clarify this point for readers
Response 5: Done
Round 2
Reviewer 1 Report
Comments and Suggestions for Authors
The authors responded to the suggestions.
Author Response
Thank you very much!